# Coupling Coordination Analysis of Ecosystem Services and Urbanization in Inner Mongolia, China

**Li Na, Yangling Zhao and Luo Guo ***

College of Life and Environmental Sciences, Minzu University of China, Beijing 100081, China
* Correspondence: guoluo@muc.edu.cn

**Abstract:** Given that ecological and environmental functions are greatly influenced by rapid urbanization, a clear understanding of the relationship between ecosystem services (ESs) and urbanization is urgently needed to improve sustainable development in Inner Mongolia. In this study, we first carried out ecosystem service valuation (ESV) using the value coefficient method. We then examined the urbanization level using a comprehensive indicator system. Finally, we applied the coupling coordination degree model to analyze the coordination relationship between ecosystem services and urbanization from 1995 to 2020 in Inner Mongolia. The results showed that there was an increase in both the urbanization level and all ecosystem services excluding climate regulation, environmental purification, and biodiversity services. The coupling coordination degree (CCD) of Inner Mongolia is not ideal, and most counties remain at a low level of coordination degree. Furthermore, spatiotemporal heterogeneity was evident in the CCD of ecosystem services and urbanization as it was higher in the center and east of the country, but lower in the north and west regions. Relevant policies should be implemented to strengthen the advantages of local ecology, encourage environmentally friendly industrialization, and promote ecologically and economically sustainable development.

**Keywords:** ecosystem services; urbanization; coupling coordination relationship; Inner Mongolia

## 1. Introduction

Since the beginning of the twenty-first century, China has undergone a significant increase in urban development. The urbanization rate of China surpassed 60% at the end of 2020 [1] and, based on Northam's classification of urbanization stages, is considered to be in the accelerated development stage [2,3]. China's rapid urbanization is not only illustrated by fast population urbanization, defined as the population transfer from rural to urban districts, but is also characterized by land and economic urbanization [4]. Urbanization can drive natural ecosystems to become human–nature combined or human-dominated ecosystems [5]. Recent rapid urbanization has brought about an increasing number of environmental problems including degradation of environmental quality, scarcity of arable land, and most importantly, the decline of ecosystem provision which has affected both the structure and function of ecosystems in regional areas, thereby destroying ESs [6]. ESV can be utilized to assess the human services potential provided by regional ecosystems and to analyze the environmental transformation induced by human disturbances. In many studies, experts have considered ESV to be a key indicator used for measuring changes in the ecological environment [7].

ESs are defined as the benefits and welfare that humans derive from ecosystems, either directly or indirectly [8]. More specifically, ecosystem services encompass provision services, regulation services, support services, and recreation and culture services [9]. The integration of provision, regulation, support, and cultural services contributes significantly to the synergistic achievements of the United Nations Sustainable Development Goals [10]. Globally, however, ESs have declined substantially over the past few decades, and it is predicted that these declines will continue in the decades to come [11]. Additionally, the

structure (for example vegetation coverage and soil quality) as well as the processes (for example animal mitigation) within ecosystems are profoundly altered [12]. Intensifying urbanization is putting increasing pressure on ecosystems. With their vital role in connecting human society to nature, ecosystem services are usually impacted by urbanization either directly or indirectly [13]. Quantifying the relationship between ecosystem services (ESs) and urbanization is of utmost importance to identify the underlying mechanisms between two variables and achieve sustainable development between man and nature [14]. In this context, scholars, policymakers, and stakeholders worldwide are closely focused on investigating the interactions and coordination mechanisms between urbanization and ESs. A solution to this common and challenging issue is urgently needed to enhance regional sustainability [15,16]. It is imperative to better understand the coupling relationship and assess the coordination and conflict between urbanization and ESs to strike a long-term balance between urbanization and ecosystem protection.

Scholars have conducted numerous studies on ESs and urbanization. Studies of ESs rely primarily on the invest model or the value scale developed by Costanza et al. [17] for simulating and exploring ecosystem services, and ESV is calculated by continually optimizing the value equivalent factor [18]. In addition, investigations on urbanization have been conducted primarily based on LUCC change [19], nighttime lighting data, urbanization rate, and single-dimensional estimates [20]. A simple linear relationship usually does not exist between ecosystem services and urbanization. The relationship may involve a correlation mechanism that is dynamic and nonlinear [21]. There are different approaches currently being applied to examine such nonlinear correlations. Wang et al. [22] analyzed the interaction mechanism between ESs and urbanization utilizing curve estimation. Other researchers have used piecewise linear regression to explain the relationship between two variables in Beijing suburbs [23]. The ecological environment and urbanization are generally interconnected. Studies have been conducted on this topic specifically for metropolitan regions of high urbanization or other administrative regions [24–26]. They are however not able to provide a coherent assessment of the coupling coordination relationship between ecosystem services and urbanization for the ecological environments of relatively fragile arid and semi-arid regions and a more precise resolution for the coordinated development between two variables in the typical arid and semi-arid regions. Nevertheless, insufficient research is being done on coupling coordination between urbanization and ESs, particularly in ecologically fragile zones, e.g., arid regions and plateaus, which play an essential role in long-term sustainable development and ecological management in these regions [27]. This study examines the spatio-temporal patterns of the coordination relationships between ESs and urbanization in the typical arid and semi-arid region using a coupling coordination degree (CCD) model. Further, scholars from different backgrounds and subjects have employed different methods and models including ordinary least square model (OLS) [28], geographical weighted regression model (GWR) [29], and panel data regression [30] model to investigate the coupling coordination relationship between socioeconomic and ecological systems. Nonetheless, these models can only explain urbanization's unidirectional effect on the natural environment and fail to take into account the spatial-temporal effect. The coupling coordination degree (CCD) model fills the gaps mentioned above [31]. Additionally, the existing literature is primarily concentrated on the regional, provincial, and municipal scales with few studies conducted at the county level in the entire autonomous region. To fill this research gap, the current study was implemented at the county level. Referring to administrative units, the evaluation results are more targeted and provide more effective guidance for actions taken by the local government [32]. Therefore, conducting evaluations at the county level is a useful way of making sustainable development a reality and improving the coupling coordination development of urbanization and ecosystem services.

As a typical resource-based area, the Inner Mongolian economy is heavily dependent on coal mining and extraction of other natural resources to promote its economic prosperity, which has led to extensive resource consumption and fragile environmental conditions in the entire region. The rapidly degrading environment in turn hinders local urbanization de-

velopment, and Inner Mongolia faces severe challenges in its attempt to achieve sustainable development. With the above considerations, we developed a comprehensive assessment indicator system that includes the multiple aspects of population, economy, and land to identify urbanization. In addition, the spatial-temporal evolution characteristics of ESs and urbanization were identified in Inner Mongolia over the period 1995 to 2020. Specifically, this article aimed to (1) investigate the spatio-temporal distribution pattern of individual ESs in Inner Mongolia from 1995 to 2020; (2) evaluate the urbanization level of Inner Mongolia from 1995 to 2020 based on three dimensions relating to population, economy, and land; and (3) utilize a CCDM model to get a clear understanding of the coordination interaction between ESs and urbanization. The results serve as a policy-making basis for achieving environmental protection as well as sustainable development in Inner Mongolia and a reference for improving the coordination relationship between ESs and urbanization in similar areas worldwide.

## 2. Study Area, Data Sources, and Research Methods

### 2.1. Study Area

Inner Mongolia stretches across Northern China covering 1.183 million km$^2$ and containing 25 million people [33]. It consists of 101 counties and occupies 12.3% of China's total land area (Figure 1). The climate in this area is largely dominated by the continental monsoon due to its high latitude and altitude and distance from the ocean [34]. The average annual temperature is 0–8 °C declining gradually from south to north. Annual precipitation decreases progressively from northeast to southwest ranging from 50 to 550 mm, with 75% occurring between July and September [35]. The area consists of six major vegetation ecological zones: shrub desert, desert steppe, forest steppe, typical steppe, meadow steppe, and deciduous forest. Xilamuren, Xilingol, Horqin, and Hulunbuir are four major prairies in the region, and Badain Jaran, Ulanbuh, Kubuqi, and Tengger are four major deserts. According to LUCC in 2020, grassland made up 46.13%of Inner Mongolia's total area, followed by desert or barren land (25.37%), forested land (14.46%), cultivated land (9.99%), wetland (3.14%), and construction land (1.02%). In recent years, Inner Mongolia has seen rapid urbanization due to its abundant energy resources, which also contributed to the region's social and economic progress. During the past few years, the region's social economic development and the construction of infrastructure have made great progress due to its energy resources advantages. The urbanization level has greatly improved, and now it has entered a phase of rapid urbanization development. Rapid urbanization not only occupies the ecological environment directly, but also exerts a coercive influence on the environment, energy, and resources. Therefore, conflicts between the ecological environment and urban development are becoming more dramatic.

### 2.2. Data Sources

In this article, LUCC datasets were obtained from the Chinese Academy of Sciences Resource Environmental Data Center (RESDC). These LUCC data with a spatial resolution of 1 km and accuracy of 90% were considered appropriate for this research [2]. Then, by combining supervision classifications and manual visual interpretations as well as field surveys, land use types were classified into grassland, forest land, cultivated land, water body, construction land, and other land (Figure 2). Additionally, population density and gross domestic product with a resolution of 1 × 1 km from 1995 to 2020 were also available on the website of RESDC. For the purpose of facilitating the exploration of the spatial distribution characteristics of Urbanization data and ESV, a grid of 2 × 2 km was applied to spatial sampling, and using ArcGIS software, the mean values in a total of 286,793 grid squares of the 6-phase data were calculated.

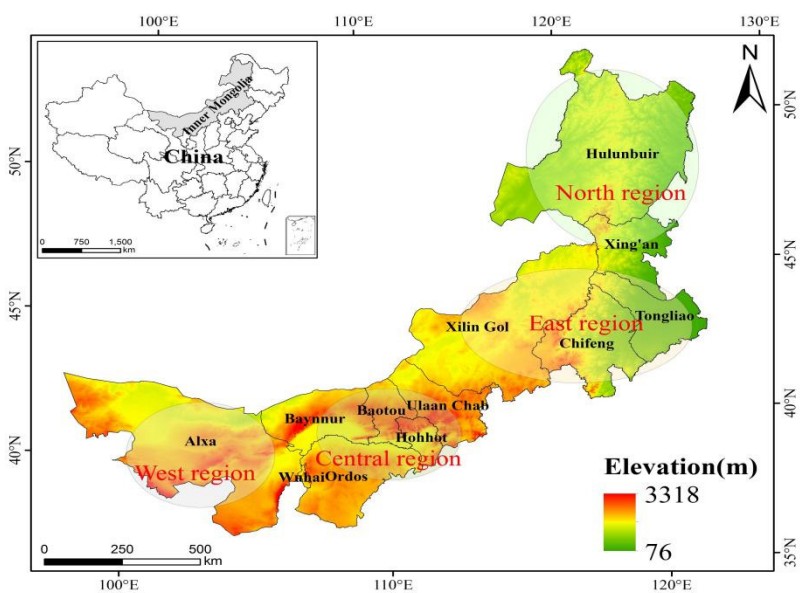

**Figure 1.** Location of the study area.

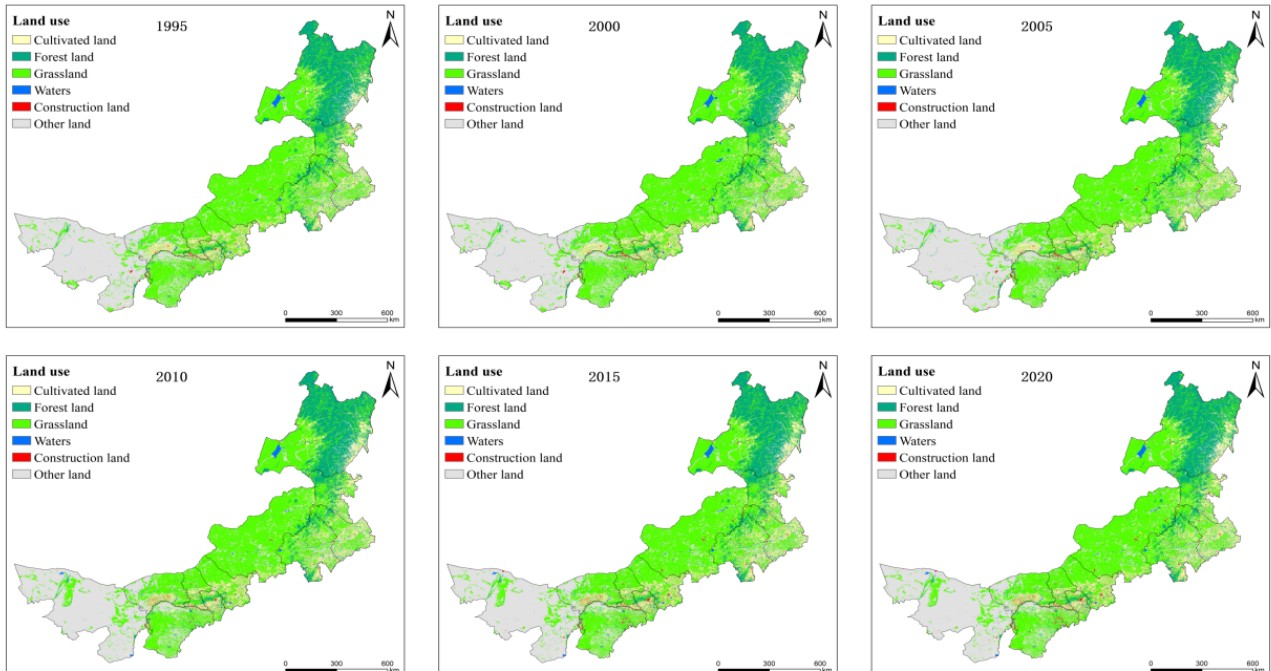

**Figure 2.** Land use types in Inner Mongolia from 1995 to 2020.

*2.3. Study Methods*

A total of 286,793 grids were generated over the six-year period (i.e., 1995, 2000, 2005, 2010, 2015, and 2020) using the "Create fishnet" function tool [36]. The spatial resolution of each fishnet cell was set to 2 × 2 km applying the ArcGIS 10.4 spatial analysis tool.

2.3.1. Assessing Ecosystem Service Values

ESV are widely evaluated by applying the equivalent factor methodology on the basis of the value per unit area [37,38]. The ESV is evaluated here according to the improved method of the equivalent factor [39], and the value equivalent factor is also adjusted according to specific circumstances. The equivalent factor of ESV according to Xie et al. [40] in a unit of measurement that is standard equals the economic value of natural grain produced annually on one hectare of farmland, equivalent to approximately one-seventh

of the national grain average value of that year the same. The inter-annual variations in grain production's economic value were considered to calculate grain value on average in the area and the adjusted ESV of each standardized equivalent factor in the research field referred to 2172.62 yuan/hm$^2$. The ESV value coefficients for each unit area are presented in Table 1. The ESV is estimated using the formula below:

$$ESV = \sum_i \sum_j A_i \times VC_{ij} \tag{1}$$

where ESV represents the total ecosystem services value, $A_i$ represents land use type $i's$ area, and $VC_{ij}$ is the land use type $i$ value coefficient and ecosystem service function type $f$.

**Table 1.** Value coefficients of ESV each unit area (yuan/hm$^2$·year).

| Ecosystem Services Functions | | Cultivated Land | Forestland | Grassland | Waters | Other Land |
|---|---|---|---|---|---|---|
| Supply services | Food production | 263.5 | 83.7 | 71.3 | 248.0 | 3.1 |
| | Raw material production | 124.0 | 195.3 | 105.4 | 71.3 | 9.3 |
| | Water supply | 6.2 | 102.3 | 58.9 | 2569.9 | 6.2 |
| Regulation services | Gas regulation | 207.7 | 641.7 | 375.1 | 238.7 | 34.1 |
| | Climate regulation | 111.6 | 1922.0 | 988.9 | 709.9 | 31.0 |
| | Environmental purification | 31.0 | 558.0 | 325.5 | 1720.5 | 96.1 |
| | Hydrological regulation | 83.7 | 1196.6 | 725.4 | 31,694.4 | 65.1 |
| Support services | Soil conservation | 319.3 | 781.2 | 455.7 | 288.3 | 40.3 |
| | Nutrient cycling | 37.2 | 58.9 | 34.1 | 21.7 | 3.1 |
| | Biodiversity | 40.3 | 713.0 | 415.4 | 790.5 | 37.2 |
| Cultural services | Aesthetic landscape | 18.6 | 310.0 | 182.9 | 585.9 | 15.5 |

### 2.3.2. Comprehensive Urbanization Indicators Assessment

Urbanization is categorized into three-dimensions, which can be further broken down into three categories: population, economy, and land urbanization [41]. Specifically, the three key metrics were the population density (POP), the GDP density (GDP), and the urban land percentage (UBP). Afterwards, the urbanization factor layers were obtained by aggregating the 1 × 1 km resolution data of *PD*, *GDPD*, and *BLP* into each 2 × 2 km evaluation grid. Finally, the comprehensive urbanization level (CUL) was obtained by superimposing *PD*, *GDPD*, and *BLP*. The formula is as follows:

$$CUL_j = \frac{1}{3}\left(PD'_j + GDPD'_j + BLP'_j\right) \tag{2}$$

where $CUL_j$ refers to the comprehensive urbanization level in unit grid $j$, and $PD'_j$, $GDPD'_j$, and $BLP'_j$ correspond to the standardized values of population density (PD), gross domestic product density (GDPD), and build-up land percentage (BLP).

### 2.3.3. Coupling Coordination Relationship between ESs and Urbanization

The coupling associations between the urbanization and ecosystem services originated from a measure that exists in physics [42–44]. It is illustrated that a positive dynamic relationship forms from interactions between two or more systems in which both systems benefit mutually [45]. Using the theory of coupling in physics, we formulate a coupling model calculation of ecosystem services and urbanization as shown below:

$$C = 2\left\{\frac{W_{UD} \times W_{ES}}{(W_{UD} + W_{ES})^2}\right\}^{\frac{1}{2}}, C \in [0,1] \tag{3}$$

where $C$ represents the coupling level of the ecosystem services and urbanization, which has the value of 0–1; $W_{UD}$ represents the urbanization assessment value; and $W_{ES}$ refers to

the comprehensive ecosystem services assessment value. Higher *C* values indicate stronger interactions between urbanization and ecosystems. The coupling degree only reflects their correlation level in contrast to the actual development status between the two systems. Possibly, the two systems have low values, but high coupling degrees. Thus, it is important to note that the high coupling value is different from the high horizontal coupling. Thus, the CCDM model was applied to measure the level of coordination between ecosystem services and urbanization using the following formulas:

$$\begin{cases} D = \sqrt{C \times T} \\ T = \alpha W_{UD} + \beta W_{ES} \end{cases} \tag{4}$$

where *D* refers to the CCD of the urbanization and ecosystem; *T* indicate the overall synergy effect represented in the two systems' comprehensive evaluation index; and *α* and *β* refer to the two systems' weight coefficients, most commonly in 0–1. The current study presumes that urbanization and the ecosystem are equally important; therefore, both *α* and *β* were given the same value of 0.5 [46].

We used the CCD division standard below to analyze the coordination development of ecosystem services and urbanization, and the divisional results are presented in Table 2 [47,48].

**Table 2.** CCD standard between ESV and urbanization.

| Range of *D* Value | Level of CCD | Type of CCD |
|---|---|---|
| $0.8 < D \leq 1$ | Quality coordination (L1) | $W_{UD} > W_{ES}$ Quality coordination with lagging ecosystem<br>$W_{UD} \approx W_{ES}$ Quality coordination of urbanization and ecosystem<br>$W_{UD} < W_{ES}$ Quality coordination with lagging urbanization |
| $0.6 < D \leq 0.8$ | Intermediate coordination (L2) | $W_{UD} > W_{ES}$ Intermediate coordination with lagging ecosystem<br>$W_{UD} \approx W_{ES}$ Intermediate coordination of urbanization and ecosystem<br>$W_{UD} < W_{ES}$ Intermediate coordination with lagging urbanization |
| $0.5 < D \leq 0.6$ | Primary coordination (L3) | $W_{UD} > W_{ES}$ Primary coordination with lagging ecosystem<br>$W_{UD} \approx W_{ES}$ Primary coordination of urbanization and ecosystem<br>$W_{UD} < W_{ES}$ Primary coordination with lagging urbanization |
| $0.4 < D \leq 0.5$ | Basic incoordination (L4) | $W_{UD} > W_{ES}$ Basic incoordination with hindered ecosystem<br>$W_{UD} \approx W_{ES}$ Basic incoordination of urbanization and ecosystem<br>$W_{UD} < W_{ES}$ Basic incoordination with hindered urbanization |
| $0.2 < D \leq 0.4$ | Intermediate incoordination (L5) | $W_{UD} > W_{ES}$ Intermediate incoordination with hindered ecosystem<br>$W_{UD} \approx W_{ES}$ Intermediate incoordination of urbanization and ecosystem<br>$W_{UD} < W_{ES}$ Intermediate incoordination with hindered urbanization |
| $0 < D \leq 0.2$ | Extreme incoordination (L6) | $W_{UD} > W_{ES}$ Extreme incoordination with hindered ecosystem<br>$W_{UD} \approx W_{ES}$ Extreme incoordination of urbanization and ecosystem<br>$W_{UD} < W_{ES}$ Extreme incoordination with hindered urbanization |

## 3. Results

### 3.1. Spatiotemporal Change of the ESV

The distribution pattern of individual ESs in Inner Mongolia showed significant spatial heterogeneity from 1995 to 2020 (Figure 3). Natural conditions across the study area varied greatly, and districts varied significantly in their ability to provide ecosystem services. The spatial pattern of water supply services in most districts has remained mostly unchanged over the past 25 years. WY displayed the highest high–low spatial distribution pattern attributed to regional differences in precipitation from the northeast to the southwest. The most significant changes in WY were observed in the central region between 2000 and 2015, with initial decreases followed by increases, mainly due to precipitation and vegetation evapotranspiration. SR values were lower in the western region, where bare and unused land make up most of the land use type. Higher SR districts were concentrated in the area with abundant precipitation and dense vegetation coverage, making it evident that

forests contributed significantly to soil retention, alleviating urbanization's negative effect on soil retention. The spatial pattern of SR in the west has changed dramatically since 2000. Climate regulation services are located high in the northeastern region. It becomes clear that high-value areas are distributed in the same way as forest types when superimposed on the land use map, indicating that this area is highly climate-regulating. Overall, climate regulation value decreased during the study period. FP is influenced by a variety of factors, including the amount of farmland. The improvement of agrotechnology contributes to an average FP increase of 1.76% despite the GFGP implementation. Recreation potential did not show significant changes over the studied time interval. Nevertheless, this service displayed clear spatial heterogeneity as high recreation values were mainly found near water sources and mountains.

The spatial pattern of ESV in Inner Mongolia was relatively stable, but there was an evident spatial distribution pattern owing to differences in land use structure and geographic location (Figure 4). Forest coverage in hilly and mountainous areas in northeastern Inner Mongolia contributed to increased ESV. ESV in the west region was low on account of construction land expansion under rapid urbanization and barren land distribution in this area. Farmland and grassland mainly dominate the medium-ESV regions. In general, ES was spatially distributed with the highest values in forests and wetlands and the lowest values in urban areas. Total ESV had a small, fluctuating upward trend from 1995 to 2020. It increased from nearly 384.27 billion in 1995 to 386.3.7 billion in 2020, indicating a slight improvement of approximately 0.55% in ecosystem service functions (Table 3). Across eleven ecosystem service categories, regulation services accounted for the greatest percentage of contributions with 67.27% of ESV in 1995 and 67.35% in 2020. Individual ecosystem services such as environmental purification and biodiversity decreased. The climate regulation service was the most severely impacted regulatory service, with its value decreasing by 0.17% over the 25 years from 1995 to 2020. The total ESV increased by approximately 2.95 billion yuan due to increased hydrological regulation.

**Table 3.** ESV of different ecosystem services in Inner Mongolia ($10^8$ yuan).

| Ecosystem Services Functions | ESV | | | | | |
|---|---|---|---|---|---|---|
| | 1995 | 2000 | 2005 | 2010 | 2015 | 2020 |
| Food production | 84.41 | 85.64 | 85.45 | 85.92 | 85.93 | 85.89 |
| Raw material production | 105.37 | 105.21 | 105.00 | 105.87 | 105.83 | 105.77 |
| Water supply | 86.29 | 87.26 | 84.81 | 86.49 | 87.47 | 87.78 |
| Gas regulation | 340.87 | 339.01 | 338.03 | 341.41 | 341.30 | 341.13 |
| Climate regulation | 872.14 | 864.00 | 860.91 | 870.96 | 870.99 | 870.63 |
| Environmental purification | 322.08 | 320.39 | 318.23 | 321.42 | 321.78 | 321.83 |
| Hydrological regulation | 1049.90 | 1062.09 | 1031.89 | 1052.72 | 1064.77 | 1068.61 |
| Soil conservation | 421.26 | 419.41 | 418.28 | 422.29 | 422.16 | 421.94 |
| Nutrient cycling | 33.07 | 33.01 | 32.94 | 33.22 | 33.21 | 33.189 |
| Biodiversity | 364.43 | 361.50 | 359.76 | 363.99 | 364.12 | 364.03 |
| Aesthetic landscape | 162.90 | 161.77 | 160.79 | 162.76 | 162.91 | 162.90 |
| Total | 3842.73 | 3839.30 | 3796.11 | 3847.07 | 3860.47 | 3863.71 |

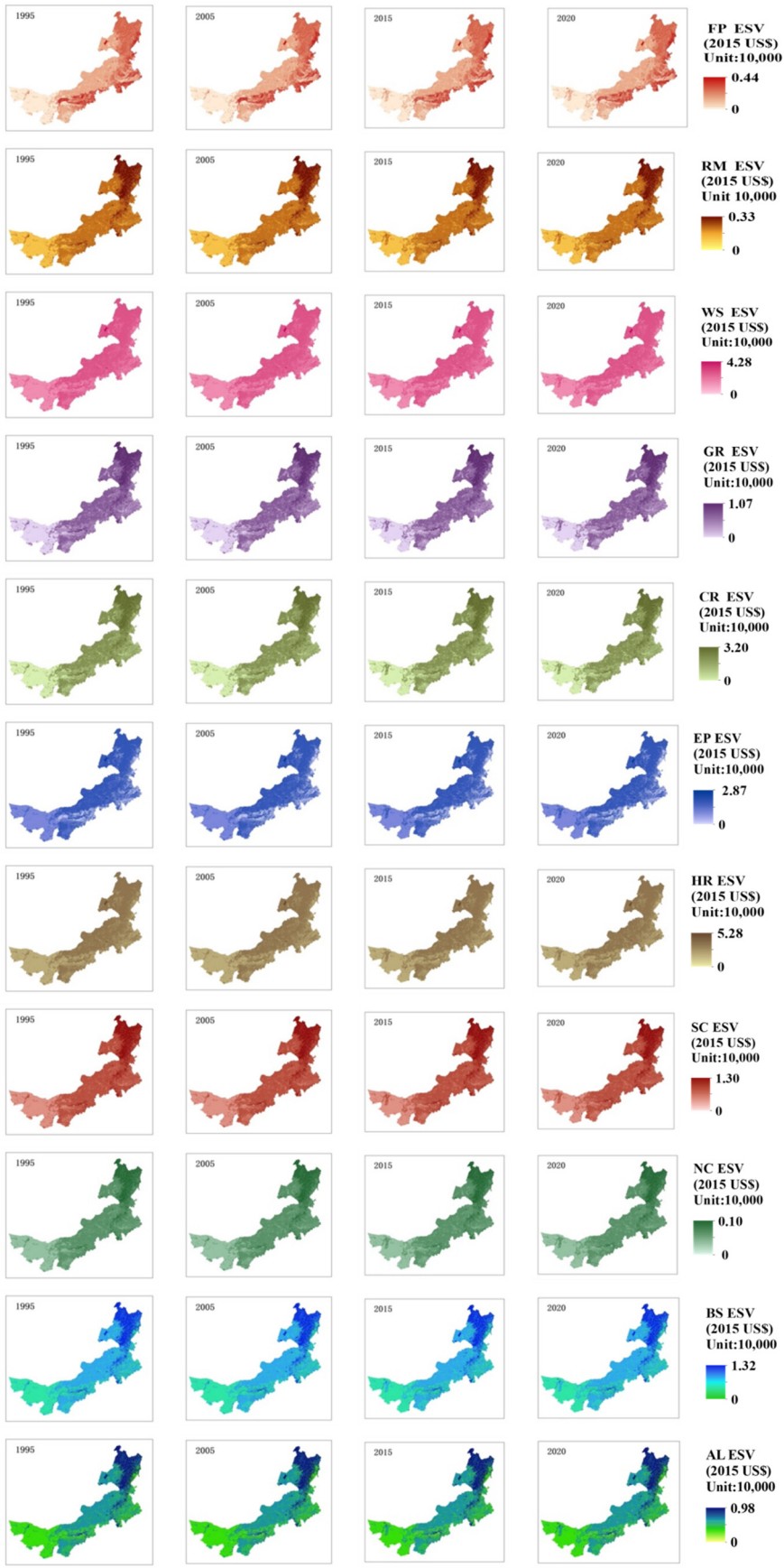

**Figure 3.** Spatial pattern of individual ESV in Inner Mongolia from 1995 to 2020.

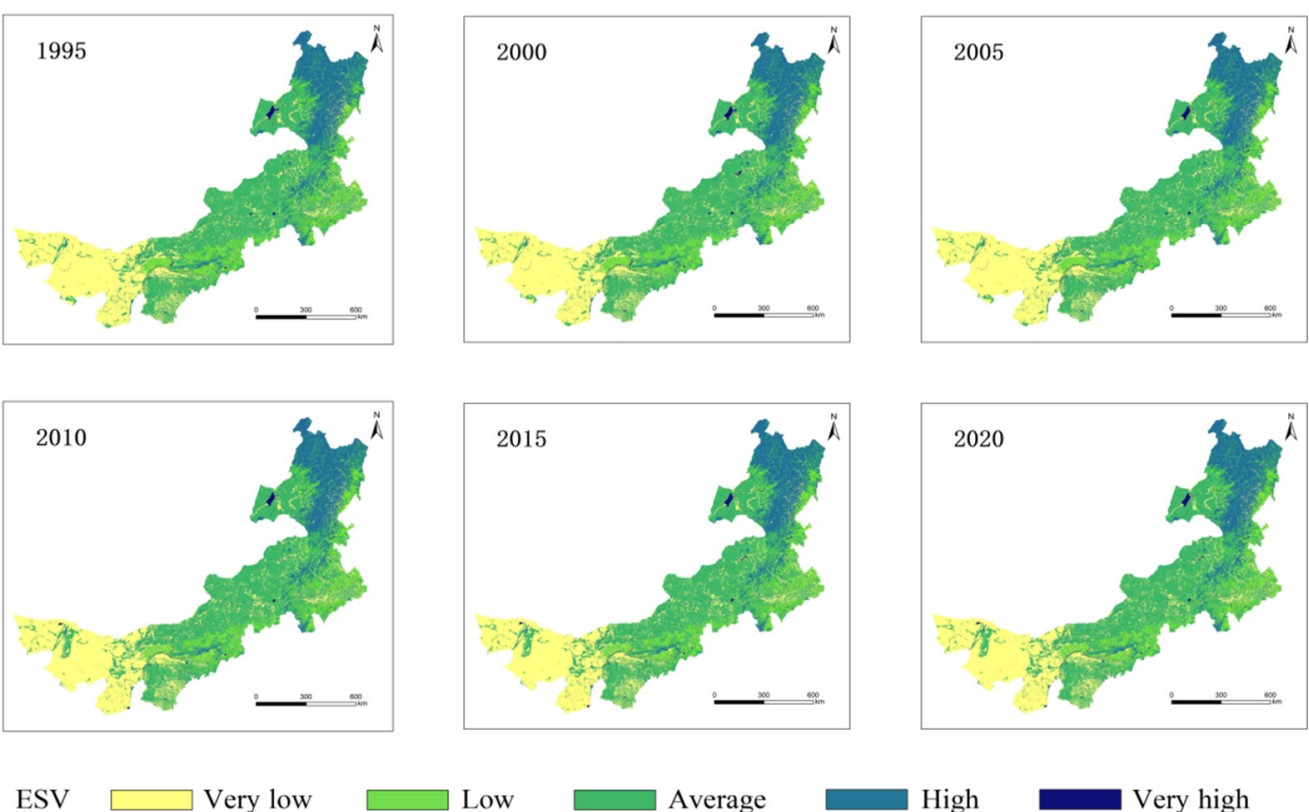

**Figure 4.** Spatial pattern of total ESV in Inner Mongolia from 1995 to 2020.

*3.2. The Dynamics of Urbanization in Inner Mongolia*

The comprehensive evaluation of Inner Mongolia's urbanization level in this study was conducted using economic, population, and land urbanization factors. The spatial pattern of urbanization in Inner Mongolia from 1995 to 2020 has experience rapid growth, especially since 2000, as shown in Figure 5. The urbanization level of all districts in Inner Mongolia increased from 1995 to 2020 despite regional variations. The Hubao-E urban agglomeration saw the greatest increase among all districts. The Hubao-E urban agglomeration had the highest level of urbanization with a maximum value of 0.65 in 1995. There were many areas of the region with low levels of urbanization, such as Hulun Buir, Xilin Gol, and Alxa (CUB: 0–0.12). Since the grasslands and mountains of Northern Inner Mongolia create an ecological barrier, many ecological restoration projects have been implemented there, resulting in a relatively low level of urbanization development in this region. Hubao-E had the highest urbanization score, still higher than peripheral areas in 2015. The urbanization level in Inner Mongolia progressively decreased from the urban center to peripheral areas. In other words, the highest level of urbanization was found in the extended urban function zone, following the new urbanization zone, with the lowest level occurring in the ecological conservation zone.

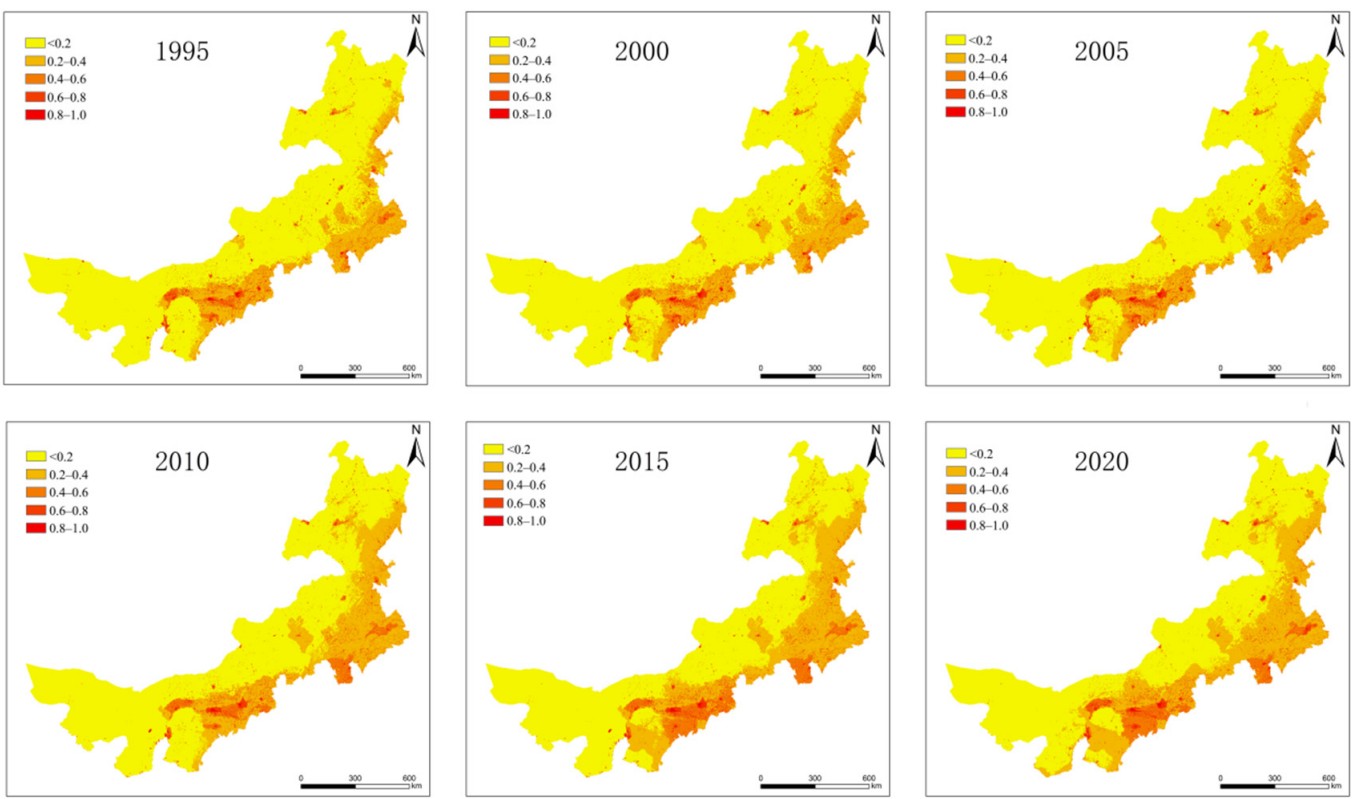

**Figure 5.** Spatial distribution of urbanization from 1995 to 2020 in Inner Mongolia.

### *3.3. CCD between Urbanization and Ecosystem Services*

### 3.3.1. Coupling Degree of ESs and Urbanization

Figure 6 shows the CD of ESs and urbanization in different counties of Inner Mongolia from 1995 to 2020. The CD consists mainly of four categories: low-level coupling, antagonism, running-in, and high-level coupling. The coupling degree across Inner Mongolia shows a clear spatial distribution and wide differences among counties. In 1995, approximately 38% of the counties exhibited a low-level coupling between ESs and urbanization. High-level coupling and running-in coupling between ESs and urbanization occurred in 48% and 14% of the counties, respectively. By 2020, 4% of the counties had transitioned to high-level coupling from the running-in coupling. The coupling degree in the Horqin Right Middle Banner decreased from the running-in to low-level from 1995 to 2010 and has been maintained at running-in coupling levels since 2010. Ecosystem service provision exceeded the urbanization level of Ejin Horo Banner after 2005, and the gap between the two progressively increased, leading to an increase in its coupling degree. The Horqin District coupling degree continually declined. The coupling degrees of Bairin Youqi, Hainan District, Tumote Right Banner, and Dongsheng District steadily increased, while in Alxa Zuoqi, it decreased from 1995 to 2020. The coupling degrees of other counties in Inner Mongolia slightly changed but remained at the same coupling stage between 1995 and 2020.

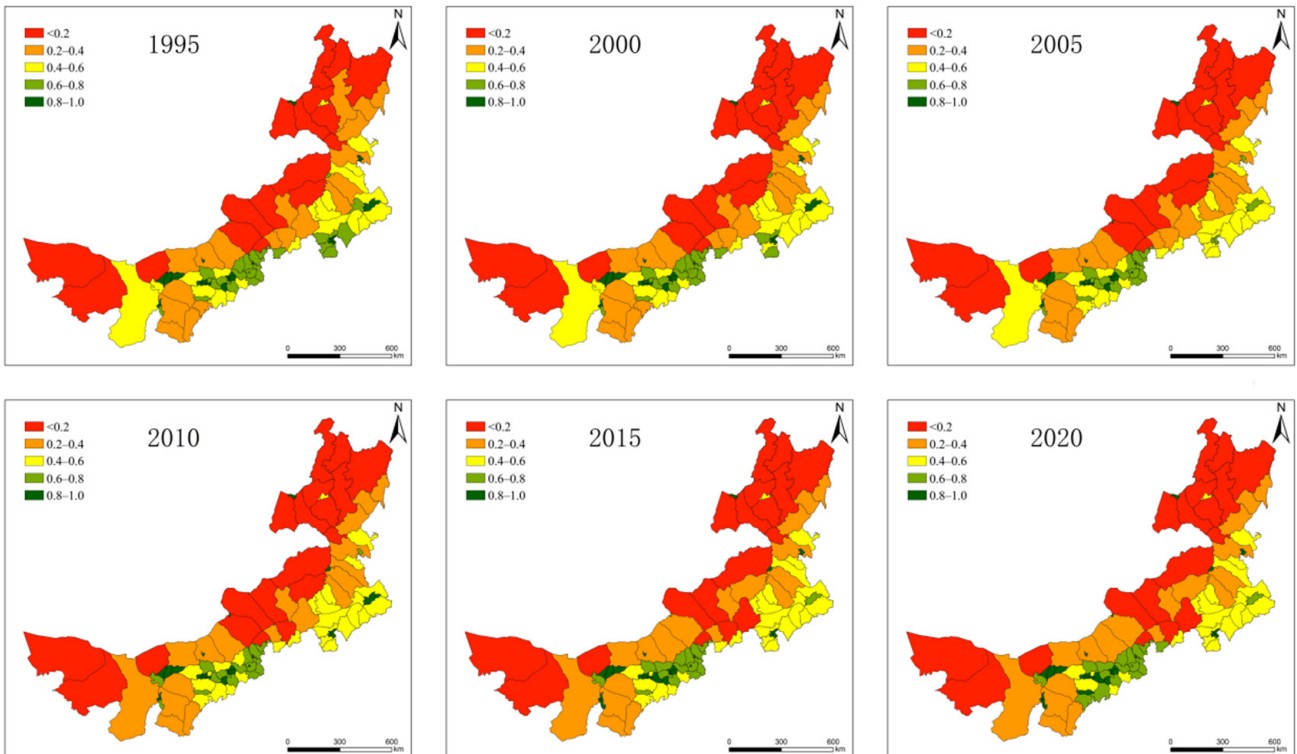

**Figure 6.** Coupling degree of ESs and urbanization in Inner Mongolia from 1995 to 2020.

### 3.3.2. Coupling Coordination Analysis of Urbanization and Ecosystem Services

The spatial distribution of CCD between ecosystem services and urbanization on a county level in Inner Mongolia was analyzed using ArcGIS software (Figure 7). Regarding the spatial pattern, the coordinative relationship performed better in the central urban agglomeration, namely Baotou, Hohhot, and Ordos counties, which is consistently in the primary or quality coordination development stage. In recent years, Ordos and Baotou faced industrial transformation and labor shortages, actively developed tourism as a means of improving their industrial structure, and invested in technological innovation to increase enterprise productivity and decrease environmental pollution produced by traditional industries [49]. In contrast, the cities located in the east region including Tongliao, Chifeng, and Xilin Gol have not yet achieved a coordination relationship between ecosystem services and urbanization, remaining in the basic or extreme incoordination phase. This is attributed to the large amount of grassland in these cities, as well as their rapid urban development in recent years. Intensive human activities have negatively affected ecosystem structure, process, and function and reduced the supply of ESs, resulting in incoordination between the two variables. The overall CCD in the western zones (Alxa) was unchanged, remaining at the extreme incoordination level. In the north region, containing the cities of Hulunbuir and Xing'an, the coordination relationship between ecosystem services and urbanization failed to perform well, remaining in the intermediate incoordination stage. In conclusion, urbanization in Inner Mongolia significantly impacted the ecosystem during the research period. Aside from a few counties, the coordination relationship between ecosystem services and urbanization was unfavorable, and there was a significant regional difference with a higher value in the northeastern than the southwestern parts.

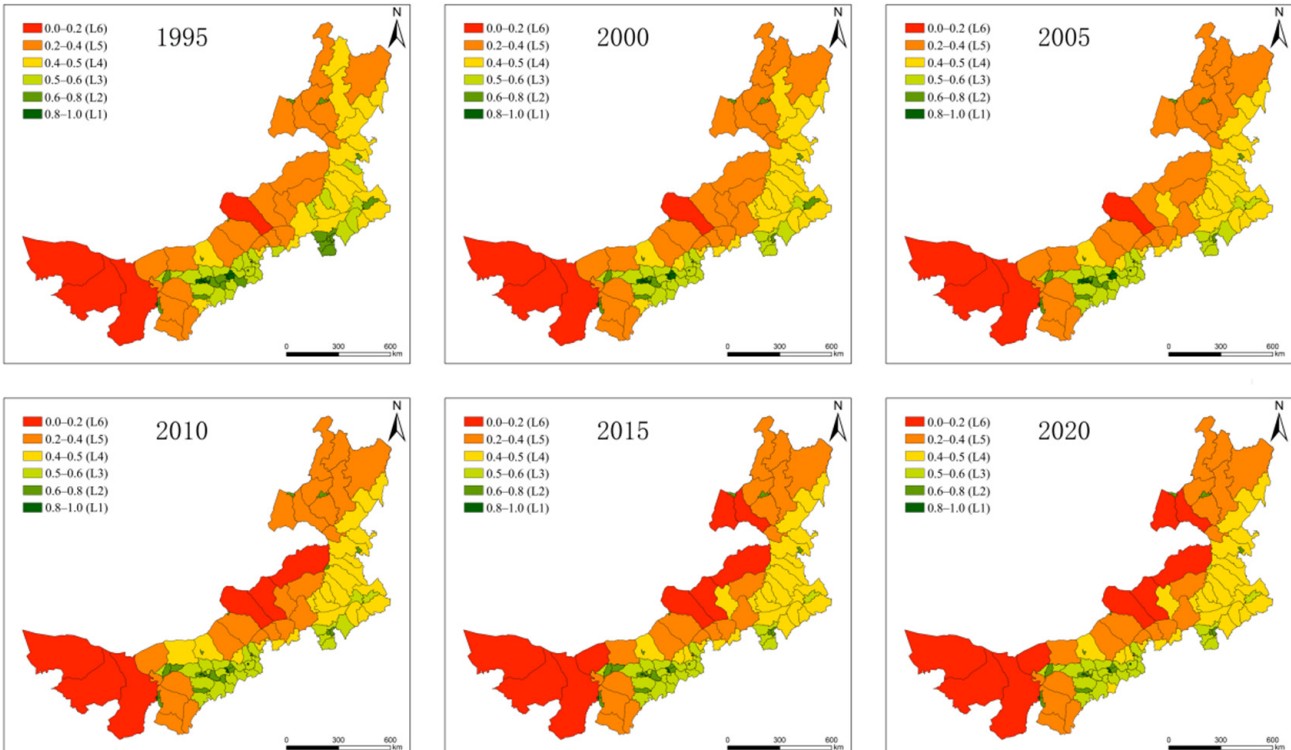

**Figure 7.** Spatial pattern of CCD in Inner Mongolia.

Figure 8 illustrates the average coupling coordination value of temporal scale variations in 101 counties from 1995 to 2020, which shows the coordination level of each county except for Alxa county. Since Alxa's temporal changes from 1995 to 2020 are not significant, we excluded them from Figure 8. The CCD values in approximately 66% of the counties exhibit a fluctuating downward trend. Located primarily in city centers and surrounding areas, these counties are mostly concentrated in the regions with the highest level of economic development and urbanization. Because of rapid industrialization and urbanization, land use patterns are largely disordered (construction land increases, and forestland and grassland decrease) and natural ecosystem foundations are changed, which adversely impacts the supply of core ESs. In contrast, the CCD changes in 23% of the counties showed an upward trend, including Chahar Right Front Banner, Dengkou, Dongsheng, Dong Ujimqin, Erenhot, Haibowan District, Hangjinhouqi, Hangjinqi, Hongshan District, Hollingol, Jining District, Linhe District, Sunite Right Banner, Wuda District, Urad Middle Qi, Wushenqi, Wuyuan Banner, West Ujimqin, Xilinhot, Ejin Horo, Yuquan District, and Jungar. The Grain for Green Program and the Natural Forest Conservation Program have contributed greatly to this growth. The implementation of these projects has considerably increased vegetation coverage, improving the CCD. The remaining 11% of counties, including Darhan Muminggan, Xin Barag Youqi in Hulunbuier, and Jarud in Xingan League, had no clear changes. Urbanization and ecosystem services coexist well in these counties on account of the low land use intensity, the small influence of human disturbances, and the relatively dense vegetation coverage.

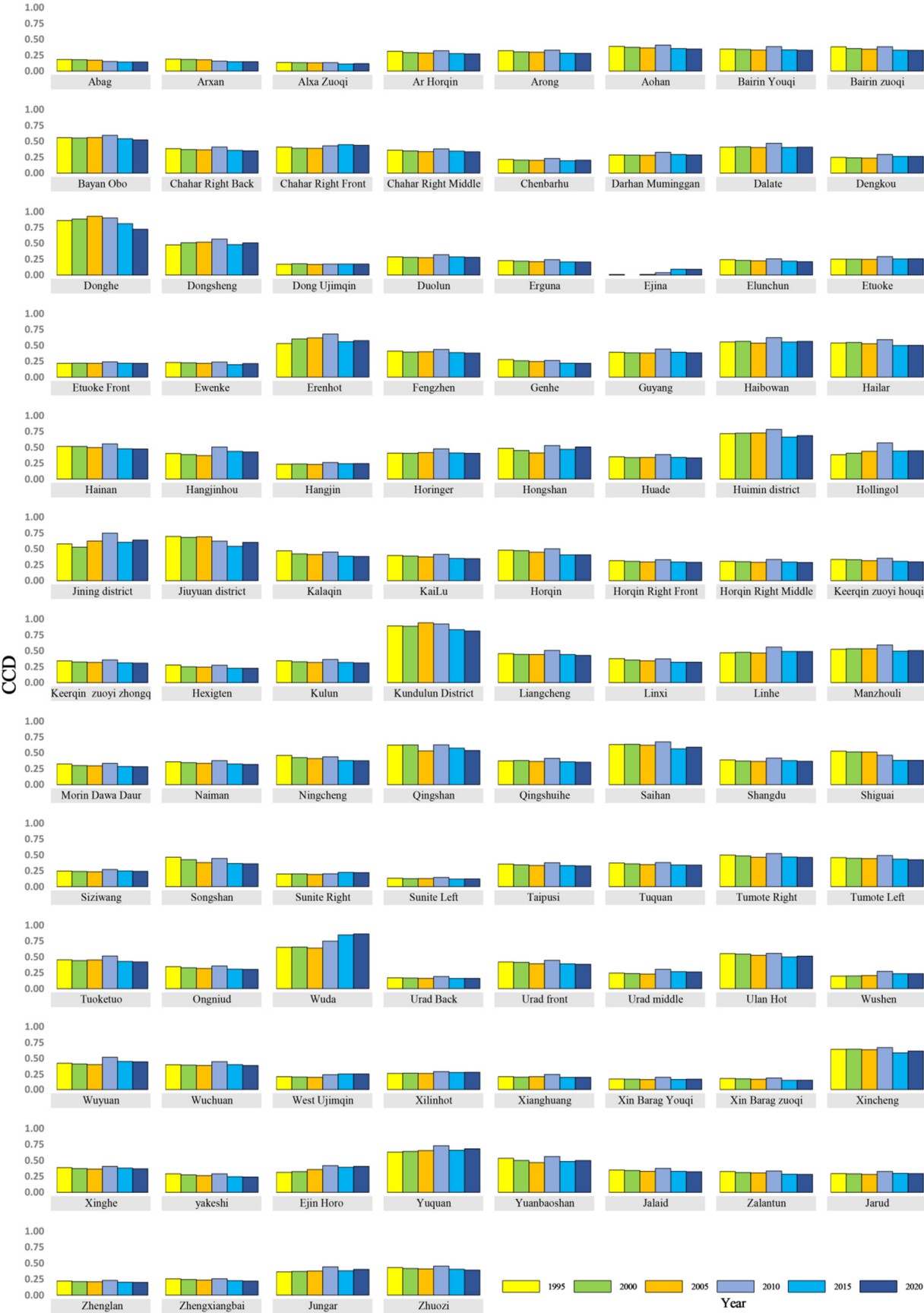

**Figure 8.** The temporal changes of the CCD between ESs and urbanization by county.

## 4. Discussion

### 4.1. Coupling Coordination Analysis between ESs and Urbanization

Inner Mongolia is a large arid and semi-arid region and a significant part of China's energy and food production base. The region has undergone dramatic changes in both socioeconomic and ecological environments during the research period. Urbanization in Inner Mongolia has grown rapidly: The Inner Mongolian urban population increased from approximately 8 million in the late 1980s to 15.14 million by 2015, and built-up space as a percentage of the provincial land area grew from 0.1% to 0.45% [33]. Growing prosperity came at an enormous cost to the environment. Generally, urbanization can negatively impact the ecological environment through economic development, population growth, traffic expansions, and energy consumption [50]. In contrast, the ecological environment can hinder urbanization development in many ways, including capital competition, population expulsion, policy interventions, and capital exclusion [51]. The rapid urbanization intensified the loss of ecosystem services, which adversely affected the ecological environment [52]. Thus, it is imperative to coordinate the relationship between the two factors reasonably and ensure the long-term development and sustainability of arid and semi-arid regions. If the contradiction between two variables decreases and both move in a positive direction, the CCD value will increase.

High-value ecosystem services are predominantly distributed in environmentally protected areas and low-urbanized areas, whereas low-value ecosystem services are mainly located in highly urbanized areas. Urbanization interacts with the benefits derived from land use types, which supports the findings of Zhou et al. [53]. Hulunbuir has the highest value of ecosystem services owing to the rich forest coverage, but coupled situation between ESV and urbanization has not yet reached a balanced and coordinated stage because of the low urbanization development. Chifeng City, as a highly urbanized region, expands its built-up land by occupying other types of arable land, which has resulted in a corresponding decline in ecosystem service value. Therefore, an increase in built-up land area per unit can negatively affects the total value of ecosystem services. In accordance with Tian's findings, a higher level of urbanization leads to greater pressure on ESV [54]. There are some studies suggesting that population density may also relate to this phenomenon. Our results further show that most counties' ecosystem services-urbanization CCDs are at the preliminary incoordination stage. While some counties are at the intermediate coordination stage, only a few were either not coordinated or at the quality coordination stage. The majority of Inner Mongolia counties were in the coordinate development ecological–economic coupling transition stage and the coordination relationship between ESs and urbanization in Inner Mongolia overall failed to reach a satisfactory level. Moreover, there were noticeable distinctions between different counties, with an evident spatial distribution pattern of lower CCD values in the northeast and west, with higher values in the center. Most counties with urban agglomeration were primarily in the quality, intermediate, and primary coordination stages, particularly the counties of Ordos, Baotou, and Hohhot. Despite having higher levels of urbanization, these counties still realized coordination development between ESs and urbanization. Hohhot, Baotou, and Ordos, as part of the Golden Triangle, have become the economic and social hubs of the autonomous region owing to their rich mineral resources, excellent geographical conditions, unique location advantages, and good industry matching. Recently, responding to central and local policies, these regions have an increased focus on green development, which is defined as a gradual shift from traditional regional development modes to a sustainable development mode. This development is in accordance with the global acceptance of reasonable protection and orderly utilization of limited resources. However, the CCD level was not high in most prefecture-level cities in Inner Mongolia, including Xilin Gol League, Hulunbuir, Ulaan Chab, Chifeng, and Tongliao, which have a high population but low per capita economic indicators. Secondary and tertiary industries are not prevalent, and the overall urbanization rate is slower than in the urban agglomeration area. Both Baynnur City and Alxa League have a low level of urbanization and development owing to their special

geographical location and harsh environmental conditions. The Xing'an League is a region rich in agricultural and animal husbandry production. Agriculture accounts for 30% of the region's GDP, and the agricultural population is high. However, many other economic indicators are at low levels, resulting in a backward urbanization process. The above analysis illustrates that Inner Mongolia is still facing the double pressure of ecological environmental protection and socioeconomic growth.

Therefore, it is necessary to adopt different urbanization and ecological environmental policies in different economic regions to promote the coordinative development between ESs and urbanization in Inner Mongolia. Furthermore, the results of this study can serve as a useful reference for coordinative development between ESs and urbanization in other arid and semi-arid areas.

### 4.2. Policy Implications

We explored the coordination relationship between ESs and urbanization and found regional differences between the center and the rest of Inner Mongolia. Generally speaking, it is imperative to realize that the relationship between ecosystem service and urbanization is a coordinated process, not an unbalanced one [55]. Efforts should be made to protect land conservation and ecosystem health to ensure the synergistic development of ecosystem services and urbanization. Specifically, it is therefore more important to put city-specific policies into practice instead of implementing one-size-fits-all ecological restoration and protection measures. Cities situated in the northern, eastern, and western parts of the region must further enhance their development infrastructure and technical personnel to turn ecological environmental benefits into benefits of regional development and achieve green and sustainable development. The most efficient approach to increasing ESV is to protect ecosystems with high ESV equivalence factors (forest lands, wetlands, water bodies) [56]. Nevertheless, the government should not expand these ecosystems blindly when setting land use policies but rather should protect the original ecological state and ensure the integrity of the existing ecosystem. Several large-scale engineering projects implemented in Inner Mongolia have contributed to improving the overall level of the ecological environment, which fosters a better coordination relationship between socioeconomic development and the ecological environment. As for the cities in the central part of Inner Mongolia, the intensity and quantity of new built-up land in urban areas with extensive human disturbance need to be under strict control, and the conversion of existing built-up land into ecological land should be promoted. Additionally, it is essential to rationally organize and optimize green spaces in cities, including water bodies and parks, to balance the ecological environment and human needs. Unreasonable resource extraction and urban construction destroy ecosystem services by disrupting the supply-and-demand balance. It is thus important for these areas to close down industries that consume resources, develop green service industries, set up nature reserves, and form a system of ecosystem service that is low-consumption, green, and low-carbon, rather than one which consumes resources.

### 4.3. Limitations and Suggestions

This article identified the coordination interaction between ESs and urbanization by using an arid and semi-arid region as a case study. However, the current study has some deficiencies that need to be taken into account in future studies. First, no validation analysis was conducted; a greater quantity of field observation data is needed to confirm the reliability of the ES evaluation. Additionally, input data resolution greatly affects the accuracy of the estimated result. The data pertinent to the research area are largely on the national scale with a resolution of 1 km, leading to limitations and uncertainties in measuring ESs. Furthermore, the developed land equivalent factor was set to 0. While ESs provided by developed land have been viewed by many studies as having no value [57], some studies have argued that entertainment, tourism, and culture are valuable ESs provided by developed land. That semi-artificial or artificial ecosystems replacing natural

ecosystems in the process of urbanization will undoubtedly result in ecosystem destruction, and this eventuality coupled with the hysteresis effect of ESs leads us to suggest setting the developed land value to 0. Lastly, the urbanization evaluation index systems were based only on three indices representing the aspects of the economy, population, and land. To obtain more reliable results, future studies should consider a more accurate and comprehensive urbanization assessment indicator system.

## 5. Conclusions

This study analyzed the spatio-temporal evolution of ESs and urbanization level in Inner Mongolia from 1995 to 2020, then utilized a CCDM model to explore the coordination relationship between the two variables. This study serves as a scientific foundation for ecological environment management and sustainable development strategies. The results indicate that (1) climate regulation, environmental purification, and biodiversity services decreased while other types of services increased. In terms of ecosystem service value categories, the regulation function has the greatest value followed by supply services and support services, and the cultural services are considered to be the lowest value function. There is evident spatial heterogeneity in the capacity of ES provision in different regions. High ESVs generally were found in the forest-rich cities located in northeastern Inner Mongolia, while medium-ESVs were found in the central and eastern plains dominated by grassland and cultivated land. ESVs in the West Region were low due to barren land distribution in this area. The above changes are mainly caused by the GFGP policy and rapid urbanization, both of which have led to vegetation coverage growth and built-up land expansion. (2) The urbanization level of Inner Mongolia had a noticeable increasing trend. Furthermore, the urbanization degree of most districts presented an upward trend, and urban expansion was concentrated in the developed city centers. An obvious hierarchy of urbanization levels exists in the study region, with the higher administrative level corresponding to the higher level of urbanization. Taking Hohhot as the core city and Baotou as the sub-core city, the spatial distribution characteristics of urbanization decrease from the center to the outside as well as from the south to the north. (3) Overall, the CCDs between ecosystem services and urbanization in most counties were too low. The CCD showed a spatial trend of higher values in the center and east, and lower values in the west and north. Relating to the spatial distribution of coupling and coordination between ecosystem services and urbanization, large areas of the Inner Mongolian Autonomous Region are still dominated by imbalances.

**Author Contributions:** Conceptualization, L.G.; methodology, L.N. and Y.Z.; formal analysis, L.N.; investigation, L.N.; data curation, L.N. and Y.Z.; writing—original draft preparation, L.N.; writing—review and editing, L.G. and L.N.; visualization, L.N.; supervision, L.G. All authors have read and agreed to the published version of the manuscript.

**Funding:** This research was funded by the Ministry of Science and Technology of the People's Republic of China (2022YFF1303001).

**Data Availability Statement:** The sources and preprocessing of data are in Sections 2.2 and 2.3. Other relevant data to support this study are available from the authors upon request.

**Conflicts of Interest:** The authors declare no conflict of interest.

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
