# Peer review of "Coupling Coordination Analysis of Ecosystem Services and Urbanization in Inner Mongolia, China"

_land, doi:10.3390/land11101870_

Round 1

Reviewer 1 Report

 This manuscript is a valuable study of the Analysis of Ecosystem Services and Ur- 2 banization in Inner Mongolia, China. The presentation is clear, the data and statistics robust, and the main section is well structured and derives logically from the results obtained. I think this work could be published in ‘land’ subject to minor modifications. The topic is of high interest, especially in the light of ongoing climate forced changes that are observed in the region.

One important point is that the  figure 8 which is not helpful in case we do not see information about that.  

I missed to understand the final conclusion in the manuscript. I highly encourage authors to re-write the final paragraphs to deliver the take home massage to readers. 

Reviewer 2 Report

The manuscript presents interesting research on recent changes in the urbanization process in Inner Mongolia and their impact through the assessment of ecosystem services.

The document presents a good structure and, from the point of view of a non-native English reader, is correctly written.

However, I believe that some things should be clarified to fit publication standards:

--Formula (2). Could you please clarify the meaning of PD, GDPD and BLP, or harmonize with other acronyms?

--3. Results-3.1. Spatiotemporal Change of the ESV. In the section, WY, SR and other Ecosystem Services values are mentioned (figure 2), but these are not found in the text or in Figure 2

--Figure 2. Could you please clarify the meaning the different ESV included in figure 2, or harmonize with other acronyms?

--Please specify if the analyzed counties are 101 (page 3, line 103) or, on the contrary, 100 (figure 8).

Reviewer 3 Report

The manuscript titled " Coupling Coordination Analysis of Ecosystem Services and Urbanization in Inner Mongolia, China " intends to investigate the spatio-temporal distribution pat- tern of individual Ecosystem Services (ESs) in Inner Mongolia from 1995 to 2020 and to evaluate the urbanization level of Inner Mongolia from 1995 to 2020 based on three dimensions relating to population, economy, and land.  Moreover, the study tried to utilize a Coupling Coordination Degree Model (CCDM) to get a clear understanding of the coordination interaction between ESs and urbanization. The study area was the inner Mongolia stretches across northern China.

The research is original; it could be characterized as novel and in my opinion important to the field, it also has an almost appropriate structure, and the language has been used well. In the meanwhile, the manuscript has a quite nice extent (about 5,370 words) and it is quite comprehensive. The tables (3) and figures (8) make the paper reflect well to the reader. For this reason, paper has a "diversity look", not only tables, not only numbers, not only words.

The title, I think, is all right. The abstract reflects well the findings of this study, and it was the appropriate length. The introduction is effective, clear, and well organized but it wasn’t introduced and put into perspective what research is negotiating. Moreover, it does not contain a clear formulation and description of the research problem. Please insert a clear description and justification of the problem the article deals with. Your literature research should be critical and more informed, rather than listing previous research. This section requires significant improvement.

For the Methodology chapter, the research conduct has been tested in several areas of the world, with comparable results and will probably be tested in others. Appropriate references to the methodology included in the already published bibliography but you can put more references, from all over the world. Do not forget, the journal “Land” is international.

The results section is good. The argument flows and is reinforced through the justification of the way elements are interpreted. But the same does not apply to the Discussion and Conclusion. It is advised to revise the Discussion and Conclusion. Both sections should be consistent in terms of Proposal, Problem statement, Results, and of course, future work (as you did). Your conclusion section does not do justice to your work. Make your key contributions, arguments, and findings clearer. You must refer to the literature and previous studies in your discussion section.

Please revise the manuscript and include more references which already exist in the bibliography. I would be much more satisfied if the number of references was slightly higher (about 25 - 30 references) and I would appreciate it if it also included data other than Asia for example America, Europe, or Australia. In this way it is documented that a method that is tested in a place with its own characteristics can be implemented in other places around the world.

I think you should increase the size, probably 1000 – 1500 words.

Please complete the left side of the first page such as Citation: Lastname, F.; Lastname, F.; Lastname, F. Title. Land 2022, 11, x. https://doi.org/10.3390/xxxxx - Academic Editor: Firstname Last-name and of course type “Publisher’s Note”, Copyright and the creative common logo with the standard image. Use the template from the journal Sustainability (https://www.mdpi.com/files/word-templates/land-template.dot).

I would prefer to enclose the reference: " Zhu, S.; Huang, J.; Zhao, Y. Coupling coordination analysis of ecosystem services and urban development of resource-based cities: A case study of Tangshan city. Ecol. Indic. 2022, 136, 108706, doi:https://doi.org/10.1016/j.ecolind.2022.108706" because the similarity of this paper is 3%, just only for a typical reason not essential.
